# Interactions of *Opuntia ficus-indica* with *Dactylopius coccus* and *D. opuntiae* (Hemiptera: Dactylopiidae) through the Study of Their Volatile Compounds

**DOI:** 10.3390/plants13070963

**Published:** 2024-03-27

**Authors:** Esteban Rodríguez-Leyva, Esperanza García-Pascual, Marco M. González-Chávez, Santiago de J. Méndez-Gallegos, Juan A. Morales-Rueda, Juan C. Posadas-Hurtado, Ángel Bravo-Vinaja, Avelina Franco-Vega

**Affiliations:** 1Colegio de Postgraduados, Campus Montecillo, Texcoco C.P. 56264, Mexico; esteban@colpos.mx; 2Colegio de Postgraduados, Campus San Luis Potosí, Salinas de Hidalgo, San Luis Potosi C.P. 78622, Mexico; forestal.egp@gmail.com (E.G.-P.); abravo@colpos.mx (Á.B.-V.); 3Facultad de Ciencias Químicas, Universidad Autónoma de San Luis Potosí, San Luis Potosi C.P. 78210, Mexico; j_c247@hotmail.com (J.C.P.-H.); avelina.franco@uaslp.mx (A.F.-V.); 4Viscoelabs, Materials Research Center, Librado Rivera 390, San Luis Potosi C.P. 78200, Mexico; ja.morales@viscoelabs.com

**Keywords:** cactus pear, phytophagous insects, terpenes, methyl salicylate, *p*-vinylguaiacol

## Abstract

*Opuntia ficus-indica* has always interacted with many phytophagous insects; two of them are *Dactylopius coccus* and *D. opuntiae*. Fine cochineal (*D. coccus*) is produced to extract carminic acid, and *D. opuntiae*, or wild cochineal, is an invasive pest of *O. ficus-indica* in more than 20 countries around the world. Despite the economic and environmental relevance of this cactus, *D. opuntiae*, and *D. coccus*, there are few studies that have explored volatile organic compounds (VOCs) derived from the plant–insect interaction. The aim of this work was to determine the VOCs produced by *D. coccus* and *D. opuntiae* and to identify different VOCs in cladodes infested by each *Dactylopius* species. The VOCs (essential oils) were obtained by hydrodistillation and identified by GC-MS. A total of 66 VOCs from both *Dactylopius* species were identified, and 125 from the Esmeralda and Rojo Pelón cultivars infested by *D. coccus* and *D. opuntiae*, respectively, were determined. Differential VOC production due to infestation by each *Dactylopius* species was also found. Some changes in methyl salicylate, terpenes such as linalool, or the alcohol *p*-vinylguaiacol were related to *Dactylopius* feeding on the cladodes of their respective cultivars. Changes in these VOCs and their probable role in plant defense mechanisms should receive more attention because this knowledge could improve *D. coccus* rearing or its inclusion in breeding programs for *D. opuntiae* control in regions where it is a key pest of *O. ficus-indica*.

## 1. Introduction

Dactylopiidae, or cochineals, is a family of scale insects that includes only the genus *Dactylopius* and 11 recognized species [1] that are endemic to North and South America [2,3]. An important characteristic of these insects is that they produce carminic acid, probably as a defense mechanism against predation [4,5,6]. All the species of the genus are considered obligate parasites of Cactacea with high host specificity, particularly for the genera *Nopalea* Salm-Dyck and *Opuntia* Miller [7].

Because of the high carminic acid concentration (~20–25%) of *Dactylopius coccus* Costa, the true cochineal, it is the only species of commercial interest for production. It is reared on *Opuntia ficus-indica* (L.) Miller, the cactus pear. Carminic acid is recognized as a natural dye with cosmetic, food, pharmaceutical, textile, and plastic applications [8]. In addition, it is currently used in biomedicine [9] and as a photosynthesizing pigment in solar cells [10]. In contrast, *Dactylopius opuntiae* Cockerell, or wild cochineal, whose carminic acid content is less than 5%, is not considered useful for obtaining this substance. Rather, it is considered the key pest of *O. ficus-indica* in commercial plantations in Mexico [11,12], where plants and insects are native [7,13]. Additionally, *D. opuntiae* is an invasive pest in at least 20 countries in America, Europe, Africa, and Asia [14,15,16], where *O. ficus-indica* was adopted or naturalized and became one of the most important cultivated cactus species in the world because of its economic, environmental, and ecological benefits [13,14,17,18].

From a scientific perspective, most *D. coccus* research has focused on the basic biology of the species and the quest to understand the mechanisms of carminic acid production and its possible physiological or ecological functions [4,19,20]. On the other hand, research on *D. opuntiae* has focused on control tactics because it is a key pest of *O. ficus-indica* [14,15,21,22]. The different cultivars of *O. ficus-indica* used as hosts of both *Dactylopius* species are likely to have particular physical and chemical characteristics, as well as volatile organic compounds (VOCs) that influence the trophic plant–insect and plant–pest–natural enemy relationship, as has been shown in other models of tritrophic interactions where volatiles cause positive or negative responses in terms of attraction and establishment of insects of the same or different species [23,24].

Volatile organic compounds (VOCs) are synthesized as products of plant metabolism, and they are emitted into the environment [25] in response to biotic complexes or abiotic stresses [23,26]. These VOCs and essential oils are released from the leaves, flowers, and fruits into the atmosphere and from the roots into the soil [27,28]. This set of volatiles, essential in the defense mechanisms of plants against herbivores or in interspecific communication [23,24,27], is called the volatilome, and its analysis is carried out by gas chromatography-mass spectrometry (GC-MS) [29]. This is a field that is continuously growing with the development of analytical and data-processing methods [30]. In this regard, some research has been carried out on VOCs of *O. ficus-indica* emanating from cladodes, flowers, fruits, and the oils of its seeds [31,32,33,34,35,36]. These studies concluded that VOC composition is a function of the geographical area, species or cultivar, plant structure, state of development, and season, among other factors. However, none of these relatively recent papers included interaction with any of the important *Dactylopius* species, nor did they relate the production of VOCs to insect infestation. To our knowledge, there is only one study that evaluated VOCs in *O. ficus-indica* cladodes uninfested and infested by *D. coccus* [37]. This study reported eight types of compounds in uninfested cladodes and nine in cladodes infested by the insect. Furthermore, no other work is known to have explored VOCs of either *Dactylopius* species.

Because plant VOCs play an important role in interactions between insects and other organisms, e.g., pathogens or predators, and parasitoids [23,24,38], as well as in the plant’s response to insect attacks [39], the objectives of this work were (1) to determine the VOCs of *D. coccus* and *D. opuntiae* feeding on *O. ficus-indica* and (2) to establish the changes in the composition and proportion of VOCs in cladodes of *O. ficus-indica* uninfested and infested with *Dactylopius*. This information could contribute to understanding the variation between cultivars of both species of insects and to exploring the potential of the biological functions that these compounds play in interspecies communication.

## 2. Results

Through essential oils, it was possible to recover and identify about 80% and 90% of the volatile organic compounds (VOCs) of *D. coccus* and *D. opuntiae*, respectively. The *Dactylopius* species had 20 VOCs in common. In addition, 12 and 34 VOCs were specifically produced by *D. coccus* and *D. opuntiae*, respectively (Figure 1). Thus, the volatilome of each species was 32 or 54 compounds, and the proportion of each compound varied greatly between species (Table 1). The VOCs belonged to eight chemical groups, of which three had the highest relative abundance. Carboxylic acids and their derivatives were the most important group, accounting for 59.28% and 78.29% of the VOC abundance for *D. coccus* and *D. opuntiae*, respectively. The second group was alcohols only for *D. coccus* (12.15%), and the third group was aldehydes with 5.8% and 7.68% of the relative abundance for *D. coccus* and *D. opuntiae*, respectively. The alkanes recovered were less than 2.5% for both species. The remaining four groups of recovered compounds (ether, terpenes, ketones, and alkenes) had less than 0.55% relative abundance per group (Table 1).

As mentioned above, the number and abundance of volatiles in each group of compounds also varied greatly in each *Dactylopius* species. For example, in the carboxylic acids and their derivatives, tetradecanoic acid was the most abundant in both species, but decanoic acid, lactic acid, and dodecanoic acid presented greater relative abundance in *D. coccus*. On the other hand, for *D. opuntiae*, 2-ethylhexanoic acid and cis-5-dodecenoic acid were detected only in this species in greater relative abundance. Hexadecanoic acid, (Z,Z)-9,12-octadecadienoic acid, (Z)-9-octadecenoic acid, and octadecanoic acid occurred in both species, but their abundance differed considerably between species; again, they were more abundant for *D. opuntiae* (Table 1).

The Esmeralda and Rojo Pelón cultivars had VOC production profiles that differed before and after *Dactylopius* infestation. In both cultivars, 28 VOCs were commonly produced and identified. In addition, 35 specific compounds were identified in Esmeralda and 19 in Rojo Pelón (Figure 2). After infestation by each *Dactylopius* species in the respective *O. ficus-indica* cultivar, a contrasting difference occurred between uninfested and infested cladodes of each cultivar (Table 2). The changes were not only in the number of VOCs but also in their abundance and variation. Sometimes they decreased, sometimes they increased, sometimes some VOCs were no longer detected, and of course there were also some *de novo* compounds (Table 2). After infestation by *D. coccus*, the Esmeralda cultivar increased the number of volatiles from 63 (uninfested) to 87, of which 48 were produced de novo and belonged to nine chemical groups. In the case of Rojo Pelón *D. opuntiae*, uninfested cladodes produced 47 VOCs, and after infestation, they decreased to 38, 13 of which were identified as de novo, belonging to seven chemical groups (Table 2, Appendix A).

Although there was an enormous variation between the number and proportion of VOCs before and after infestation, it was observed that four chemical groups maintained the highest abundance in both infested cultivars. These groups were (a) carboxylic acid and derivatives, (b) terpenes, (c) alcohols, and (d) aldehydes and their derivatives. Another group, the heterocycles, was only abundant for the uninfested Rojo Pelón cultivar (8.91%), but after *D. opuntiae* infestation, it decreased to less than 1.4%. The rest of the recovered chemical groups (ethers, ketones, aromatic derivatives, and alkanes) were less than 1.16% of the relative abundance per group in either cultivar infested by the respective *Dactylopius* species. Two of these groups (ethers and aromatic derivatives) were not detected in the infested Rojo Pelón cultivar (Table 2).

As indicated above, because of *Dactylopius* infestation in each cultivar, there were many changes in the relative abundance of compounds and the production of some de novo compounds. The de novo compounds were mostly of low relative abundance (equal to or less than 1.0%), except for some terpenes and alcohols. For example, in the uninfested Rojo Pelón cultivar, the relative abundance of terpenes was around 0.8%, but this relative abundance of terpenes changed to 15.5% after *D. opuntiae* infestation. On the other hand, the relative abundance of terpenes in the Esmeralda cultivar decreased from 18 to 13.9% due to *D. coccus* infestation (Table 2). The amount and type of terpenes were different between infested *O. ficus-indica* cultivars, but monoterpenes or their derivatives predominated in both cases (Figure 3).

The terpenes linalool oxide, trans-linalool oxide, and the alcohol 3,7,11,15-tetramethyl-2-hexadecenol reached a relative abundance of 5.06%, 5.7%, and 1.8% in the Esmeralda cultivar infested by *D. coccus*. On the other hand, the terpenes linalool, geraniol, and the alcohol 3,7,11,15-tetramethyl-2-hexadecenol registered 5.6%, 1.84%, and 3.5% of the relative abundance in the Rojo Pelón cultivar infested by *D. opuntiae*, respectively. Also, *p*-vinylguaiacol increased 2.3% in relative abundance after *D. opuntiae* infestations (Table 2).

## 3. Discussion

Previous assays of *Dactylopius* VOCs extraction, such as Headspace (HS-SPME) and extraction by Autosampler Headspace coupled to CG-MS (HS-CG-MS), did not provide the results expected for GC-MS analysis. Thus, to identify the volatiles from *Dactylopius* and its cultivar hosts, we preferred to do so using their essential oils. Essential oils were obtained by the hydrodistillation method (Appendix A), which is frequently used to obtain essential oils from plants that contain low-vapor pressure compounds or low-volatile compounds. This technique is also used for concentrating compounds with lower concentrations in the essential oil and allows working with a larger sample mass than microextraction techniques, which can potentially improve the characterization of insect VOCs [29].

In the volatilome of *D. coccus* and *D. opuntiae*, 32 and 54 VOCs were identified for each species, respectively. To our knowledge, neither of these volatilomes had been reported previously, and this may be the first contribution to this work. By their composition, these VOCs corresponded to eight different chemical groups, but there were three groups of greater abundance. These were (a) carboxylic acids and their derivatives, 59.28% and 78.29% abundance for *D. coccus* and *D. opuntiae*, respectively; (b) alcohols, which were abundant only for *D. coccus* (12%); and (c) aldehydes, 5.8% and 7.68% abundance for *D. coccus* and *D. opuntiae*, respectively (Appendix A). This composition could be one of the reasons that results were not obtained with the HS-SPME and HS-CG-MS techniques. The VOCs of *Dactylopius* species are mostly fatty acids, some of which may be part of the fat content of the insects or of the complexity of their waxy coat [40,41]. In fact, each VOC in those groups may have more than one role in structure, function, metabolism, and probably in intra- or interspecific communication. For example, *D. coccus* produces a sex pheromone [42], and *D. opuntiae* is suspected to do so as well [43]. Regarding tetradecanoic acid, which is one of the most abundant VOCs for both species of *Dactylopius*, and hexadecanoid acid, relevant to *D. opuntiae*, they have many functions in insect metabolism. One of these is to participate in the metabolic pathways of sex pheromones of some *Lepidoptera*, such as *Spodoptera lottoralis* Boisduval and *Plodia interpunctella* Hubner [44,45], but none of these compounds appear to have relevance in the pheromones of Coccoidea [46], which is the superfamily to which the Dactylopiidae belong. The methodology for identifying insect pheromones begins with live females at a particular moment of maturity and sexual behavior, and so much work remains to be carried out in order to decipher the main functions of the VOCs that turned out to be more abundant, which could lead to novel acids with shorter chains and perhaps more specific for each *Dactylopius* species.

The volatilomes of the Esmeralda and Rojo Pelón cultivars were different before and after *Dactylopius* infestation. The variation in compound production in cladodes of both cultivars prior to infestation (by *Dactylopius*) may be specific to each cultivar, as variations of other bioactive and volatile compounds have been reported in different cultivars of *O. ficus-indica* [31,35,47]. However, variation in the number and abundance of VOCs within each cultivar after infestation can be attributed to *D. coccus* or *D. opuntiae* feeding on its corresponding cultivar host, as has been demonstrated in other plants where the change in production of VOCs, particularly terpenes and sesquiterpenes, was directly associated with phytophagous insect feeding [23,24,48,49].

In the volatilomes of the Esmeralda and Rojo Pelón, before or after *Dactylopius* infestation, four chemical groups were identified as the most abundant: (a) carboxylic acid and derivatives, (b) terpenes, (c) alcohols, and (d) aldehydes and derivatives (Appendix A). The structural composition of the host, particularly the quantity of waxes, could be related to the abundance of some of these VOCs in both cultivars [47,50]. This suggestion is related to the anatomical and physiological adaptations of cacti to develop in arid environments, such as a thick and impermeable epidermis covered by a layer of waxy cuticle, a hypodermis with chollenchyma, plenty of cells with mucilage distributed in the parenchyma, and crassulaceae acid metabolism (CAM), among other characteristics [50]. Of the first and most abundant chemical groups (a), it is probable that we should mention methyl salicylate, which increased in abundance after infestation by *D. coccus* (about 5%) in the Esmeralda cultivar. The same compound was identified de novo in the Rojo Pelón cultivar infested by *D. opuntiae*, although it was low in abundance (0.3%). Methyl salicylate is a phenolic compound that has been reported to be an herbivore-induced plant volatile (HIPV) [49,51,52]. Some of these HIPVs can induce direct defense against the phytophagous insect and indirect defense by attracting their natural enemies. It is also useful for communication among plants damaged by phytophagy and others that are not yet damaged. For example, methyl salicylate emitted by plants with phytophagous mite damage was attractive to *Phytoseiulus persimilis* Athias-Henriot (Phytoseiidae) [51,52]. In the same way, it was observed that emission of this compound, after damage by psyllids in pear trees, was attractive to the predatory bug *Anthocoris nemoralis* F. (Hemiptera: Anthocoridae) [53].

In general, a slight decrease in terpene abundance (18 to 14%) was observed after *D. coccus* infestation, but a considerable increase (0.8 to 15%) occurred after *D. opuntiae* infestation. In the Esmeralda cultivar, β-linalool abundance decreased from 5.0 to 0.3%, but linalool oxide and trans-linalool oxide increased to 5.0 and 5.7%, respectively. On the other hand, in the cultivar Rojo Pelón infested by *D. opuntiae*, five de novo terpenes were identified, of which the most abundant was linalool (5.6%). Terpenes are one of the most studied groups of HIPVs, and it has been shown that some of them have a relevant role in the direct defense system against phytophages, and some volatile terpenes constitute indirect defenses of plants as they attract natural enemies such as predators and parasitoids [23,24,27,48,49].

Linalool is a monoterpene that occurs naturally in flowers and aromatic plants, but it is also produced in response to feeding by phytophagous insects, and it is part of the indirect defenses of plants [54]. For example, an increase in linalool production in tobacco plants caused by feeding Lepidoptera larvae increased the rate of egg predation and decreased the oviposition of another Lepidoptera [55]. Linalool also increased due to phytophages feeding on corn, bean, cotton, and potato plants [23], or by a zoophytophagous mirid feeding on pepper plants, and favored the action of natural enemies of their pests [49]. This can suggest that significant changes in the abundance of methyl salicylate from the above group and terpenes, particularly linalool, are probably related to each *Dactylopius* species feeding on its corresponding cultivar host.

The alcohol of greatest abundance and change was *p*-vinylguaiacol. This compound is common in plants and is part of many essential oils. In addition, it can be found in the guts of some insects, probably through the process of lignin degradation [56]. Regarding secondary plant defenses due to damage by phytophagous insects, *p*-vinylguaiacol stimulated the ovipositional behavior of the natural enemy *Coleomegilla maculata* [57], and it was also a deterrent to the oviposition of the cerambycid *Monochamus alternatus* [56]. Therefore, it is suggested that some changes in *p*-vinylguaiacol abundance may be a consequence of *Dactylopius* feeding.

In this work, 66 VOCs of both *Dactylopius* species were identified, and 125 of the Esmeralda and Rojo Pelón cladodes were infested by *D. coccus* and *D. opuntiae*, respectively. A proportion of VOCs were commonly produced in both insect species or cultivars, but others were specific to each species or cultivar (Figure 4). This is a first approach to the diversity of VOCs produced by *O. ficus-indica* and the changes that occur due to *D. coccus* and *D. opuntiae* feeding on cultivars suitable for the development of each *Dactylopius* species. More time and work will now be needed to understand the functions performed by the most relevant compounds in these interactions.

If knowledge of the interaction is improved, for example, if it is confirmed that some terpenoids favor the direct or indirect defenses of *O. ficus-indica* against *D. coccus* or *D. opuntiae*, this information could be considered in breeding programs. These programs could be aimed at improving the rearing of *D. coccus* or inducing resistance to *D. opuntiae*. In this regard, breeding programs for *O. ficus-indica* resistant to *D. opuntiae* have already been developed in Brazil and Morocco, and these have focused on physical and biochemical defense mechanisms [15,21,58]. For example, selecting cultivars with high concentrations of calcium oxalates can physically and biochemically limit phytophagous insects [59,60]. However, there are no known breeding programs for *O. ficus-indica* that consider the abundance of terpenes in cultivars and the response this can trigger in the plant’s direct or indirect defenses. This mechanism would be classified as biochemical defense, and measuring terpenes in different cultivars could improve the direction and understanding of the response.

Besides, SIMPER analysis (Appendix A) showed the components that are typical of each *Dactylopius* species and its hosts; these contribute a low percentage of each sample, so their contribution to the dissimilarity is low. This observation highlights the need to better understand the interaction between *O. ficus-indica* and *Dactylopius* because it can increase the possibilities of making proposals for sustainable management in the production of *D. coccus* or in the control of *D. opuntiae*.

## 4. Materials and Methods

### 4.1. Chemicals

The reagents used in this study were *N*, *O*-bis(trimethylsilyl) trifluoroacetamide (BSTFA), trimethylsilyl chloride (TMCS), boron trifluoride methanol solution (Sigma-Aldrich, St. Louis, MO, USA), and ethylic ether (JT Baker, Deventer, Holland).

### 4.2. Insects and Uninfested and Infested O. ficus-indica Cultivars

*Dactylopius coccus* and *Opuntia ficus-indica* Esmeralda cultivars (infested and uninfested) were originally obtained from a local provider in Jerez, Zacatecas, Mexico. *Dactylopius opuntiae* and *O. ficus-indica* Rojo Pelón cultivars (infested and uninfested) were collected from an experimental field at Colegio de Postgraduados, Campus San Luis Potosí (Salinas, SLP). These cactus pear cultivars were selected with the knowledge that each one is favorable for the development of the respective *Dactylopius* species [58]. The taxonomic identity of *Dactylopuis* species was corroborated by S. J. Méndez-Gallegos using De Lotto (1974) [40] and Ferris (1955) keys [61]. To increase material for the samples and analyses, *D. coccus* and *D. opuntiae* colonies were reared on the respective cultivars mentioned under greenhouse conditions (15 ± 2 °C, 22 ± 2 °C, and 50% RH).

### 4.3. Essential Oil of Dactylopius Species and Hosts

One hundred grams of adult females previous to the reproduction stage of *D. coccus* (80 to 85 d old) and *D. opuntiae* (30 to 35 d old) with their protective coverings (secretion substances) and 1000 g of infested and uninfested *O. ficus-indica* cladodes were used independently to obtain their essential oils by hydrodistillation. The *Dactylopius* species were manually separated from their hosts just before hydrodestillation, and the *O. ficus-indica* cladodes were cut into cubes just before hydrodestillation. The VOCs, which are components of essential oils, were obtained at boiling water temperature and extracted from the condensed water by liquid–liquid extraction with ethyl ether. The solvent was distillated, and the residual water was removed from the organic phase with anhydrous sodium sulfate. Each sample was then concentrated (to 1 mL) at 40 °C under vacuum, and the residual solvent was eliminated from each sample at atmospheric pressure at 0 °C.

### 4.4. Derivatization for Alcohol Detection

Essential oils were diluted to 2% in 500 µL heptane and introduced into a 10 mL microwave reaction tube with a gasket. Then, 100 µL of BSTFA/TMCS solution (9:1 *v*/*v*) was added to the same tube as a silanizing agent. The mixture was reacted at 90 °C under microwave irradiation (250 W microwave power) for 10 min using the Discover System 908,005 (CEM Corporation, Matthews, NC, USA) with autogenous pressure.

### 4.5. Derivatization for Aldehydes and Carboxylic Acid Detection

Essential oils were diluted to 2% in 500 µL heptane and introduced into a 10 mL microwave reaction tube with a gasket. Then, 500 µL of boron trifluoride (14% in methanol solution) was added to the same tube. The mixture reacted at 90 °C under microwave irradiation (250 W microwave power) for 10 min using the Discover System 908,005 with autogenous pressure.

### 4.6. Essential Oil GS-MS Analysis

Samples without derivatization were diluted to 2% in heptane, using 1 µL of each sample for the analysis, and each sample was analyzed in triplicate. GC-MS analysis was performed using a 7802A Network GC System coupled to a 5977E Network mass selective detector (MSD).

The separation was performed using an HP-5 capillary column (0.25 mm i.d., 30 mm, 0.25 mm film thickness) (J&W, Folsom, CA, USA). The injector was operated in splitless mode at 300 °C, with a flow of 1.0 mL/min, and the oven temperature was programmed to 40 °C for 3 min, and then heated at 3 °C/min to 300 °C with a holding time of 5 min at the final temperature. The MSD was operated at 70 eV; the ion source was set at 150 °C and the transfer line at 300 °C. VOCs were identified by interpreting their mass spectra as fragmentation in the mass range of 15 to 800 atomic mass units. The software MassHunter (Agilent B.07.01.1805, Santa Clara, CA, USA) was used for data recording. The compounds were identified by comparing the obtained mass spectra with those of reference compounds from the National Institute of Standards and Technology (NIST11) and Wiley 09. The identities of the compounds were confirmed by the Kovats retention index calculated for each peak with reference to the n-alkane standards (C7–C38) running under the same conditions.

### 4.7. Statistical Analysis

The relative percentage of each metabolite was calculated considering the peak area obtained by GC-MS of each metabolite in relation to the total area of peaks analyzed. The data represent the mean of the relative percentage of three repeats ± SD. Metabolites grouped by type for each essential oil were compared with the Mann–Whitney U test, considering the peak area of each metabolite and a *p* ≤ 0.05. The data in the graphics were expressed as the median and range of each group. GraphPad Prism 5 was used to perform the analysis. Venn diagrams were constructed using an online tool (http://jvenn.toulouse.inra.fr/app/example.html, accessed on 23 November 2023) [62]. PAST statistical software (version 4.09) was used to perform the SIMPER analysis [63].

## 5. Conclusions

This work presents an approach to better understanding the interaction between *O. ficus-indica*, *D. coccus*, and *D. opuntiae* by identifying volatile compounds in their essential oils. The abundance and proportion of VOCs of *D. coccus* and *D. opuntiae* were determined in the Esmeralda and Rojo Pelón cultivars, viable for the development of each insect species, respectively. Differential VOC production due to infestation by each *Dactylopius* species in each cultivar was also identified. Changes in methyl salicylate, terpenes, and *p*-vinylguaiacol and their likely role in plant defense mechanisms should receive more attention because they could contribute to the development of proposals to improve *D. coccus* rearing or for the control of *D. opuntiae* in those regions of the world where it is a key pest of *O. ficus-indica*.

## Figures and Tables

**Figure 1 plants-13-00963-f001:**
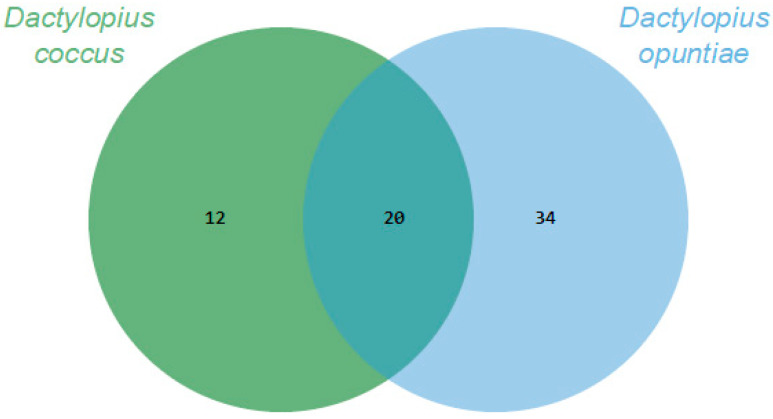
Comparison of the *Dactylopius* volatilomes using Venn diagrams based on the number of Volatile organic compounds (VOCs) obtained through essential oils for each *Dactylopius* species.

**Figure 2 plants-13-00963-f002:**
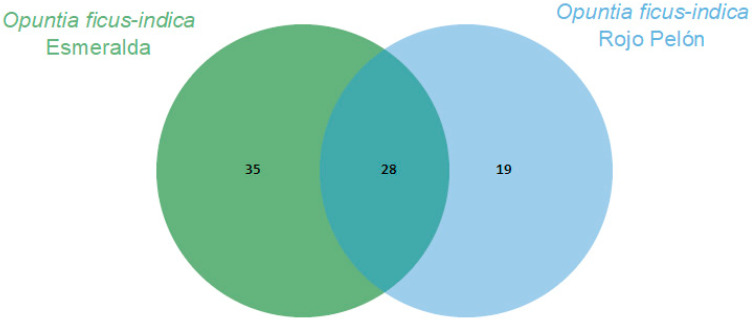
Comparison of volatilomes of the *Opuntia ficus-indica* uninfested cladodes of each cultivar using a Venn diagram, based on the number of VOCs obtained through essential oils.

**Figure 3 plants-13-00963-f003:**
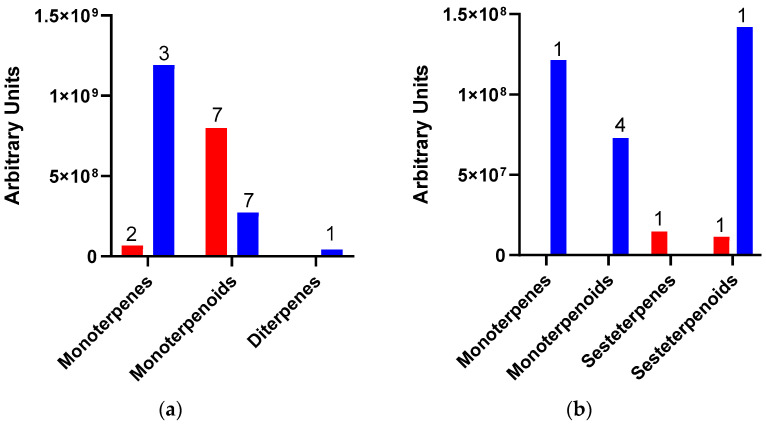
Amount and type of terpenes released by *Opuntia ficus-indica* (OFI) after *Dactylopius* infestation. (**a**) OFI Esmeralda-*D. coccus*; (**b**) OFI Rojo Pelón-*D. opuntiae*. The red columns represent uninfested cladodes, and the blue columns represent cladodes infested by each *Dactylopius* species. Data are presented as means of the peak area of each terpene (grouped by type and number of compounds).

**Figure 4 plants-13-00963-f004:**
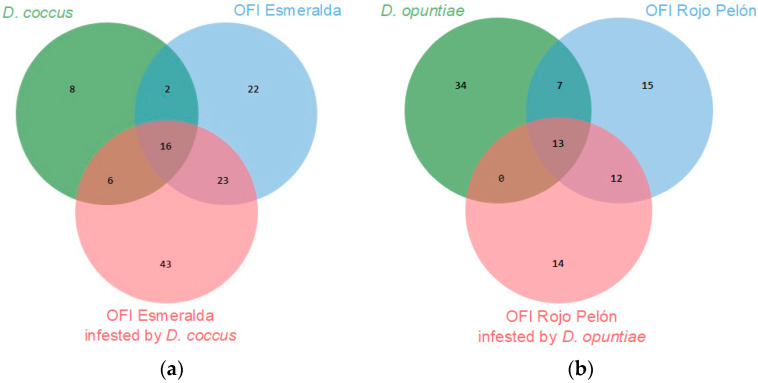
Number of VOCs, obtained through its essential, identified as common or de novo compounds between uninfested and infested *O. ficus-indica* (OFI) cultivars, (**a**) by *Dactylopius coccus*, (**b**) by *Dactylopius opuntiae*.

**Table 1 plants-13-00963-t001:** Volatile organic compounds (VOCs) obtained through the essential oils of each *Dactylopius* species.

No.	Compounds	*D. coccus*		*D. opuntiae*		
		RA (%)	KI Exp	RA (%)	KI Exp	KI Ref
*Carboxylic acids and derivatives*		*59.28%*		*78.29%*		
1	Hexanoic acid			1.09 ± 0.08		903
2	2-methylhexanoic acid			0.54 ± 0.01		950
3	Heptanoic acid	0.20 ± 0.05	1021	0.64 ± 0.16	1021	1005
4	2,4-dimethylhexanoic acid			0.37 ± 0.02		1015
5	2-Ethylhexanoic acid			10.90 ± 0.21	1036	1031
6	Lactic acid	5.58 ± 0.58	1061	0.86 ± 0.03	1062	1057
7	Glycolic acid			0.11 ± 0.03	1075	1072
8	2,6-dimethylheptanoic acid	0.61 ± 0.21	1087			
9	Octanoic acid	1.90 ± 0.74	1108	2.20 ± 0.24	1108	1108
10	2,3,4-Trimethylpentanoic acid			0.44 ± 0.02	1127	
11	Ethyl benzoate			0.12 ± 0.01	1153	1141
12	Ethyl octanoate			0.05 ± 1.77	1188	1175
13	Nonanoic acid	2.32 ± 0.64	1214	3.57 ± 0.37	1214	1205
14	2,4-dimethylnonanoic acid	0.31 ± 0.01	1234			
15	Benzoic acid	2.04 ± 0.00	1235	1.04 ± 0.01	1235	1232
16	Ethyl nonanoate	0.26 ± 0.03	1290			1282
17	2-Decenoic acid			0.42 ± 0.02	1310	1290
18	Decanoic acid	8.76 ± 2.53	1316	2.38 ± 0.12	1316	1309
19	Butanedioic acid			0.11 ± 0.03	1318	1314
20	(Z)-4-*tert*-butylcyclohexyl acetate	0.80 ± 0.02	1358	0.12 ± 0.03	1356	1346
21	Ethyl decanoate			0.18 ± 0.02	1388	1382
22	Undecanoic acid			0.63 ± 0.02	1414	1410
23	*cis*-5-Dodecenoic acid			3.18 ± 0.04	1504	
24	Dodecanoic acid	4.93 ± 0.90	1512			1509
25	Nonanedioic acid			0.40 ± 0.03	1535	1511
26	Ethyl dodecanoate	1.06 ± 0.07	1580			1566
27	Tridecanoic acid			0.49 ± 0.04	1606	1606
28	*p*-Hydroxybenzoic acid	0.24 ± 0.00	1615	0.05 ± 0.01	1616	1621
29	Hexyl salicylate	1.05 ± 0.01	1660	0.57 ± 0.02	1658	1684
30	(Z)-9-Tetradecenoic acid	0.42 ± 0.05	1702	0.22 ± 0.09	1707	1691
31	Tetradecanoic acid	21.25 ± 2.90	1717	30.15 ± 1.35	1718	1713
32	Ethyl tetradecanoate			0.40 ± 0.01	1793	1782
33	(Z)-9-Hexadecenoic acid			0.56 ± 0.12	1909	1885
34	Hexadecanoic acid	2.30 ± 1.83	1934	5.89 ± 0.42	1934	1909
35	Ethyl hexadecanoate			0.23 ± 0.01	1974	1968
36	Heptadecanoic acid			1.03 ± 0.06	2039	2009
37	(Z,Z) 9,12-Octadecadienoic acid	0.83 ± 0.73	2105	3.76 ± 0.67	2105	2087
38	(Z)-9-Octadecenoic acid	0.82 ± 1.11	2112	2.44 ± 0.33	2112	2088
39	Octadecanoic acid	1.69 ± 2.32	2139	2.58 ± 0.22	2140	2133
40	Ethyl octadecanoate	0.33 ± 0.03	2208			2181
41	Dehydroabietic acid	1.58 ± 0.00	2375	0.12 ± 0.02	2376	2385
*Aldehydes*		*5.80*		*7.68*		
42	Hexanal			1.79 ± 0.36		964
43	Heptanal			1.24 ± 0.11	1066	1069
44	Octanal			0.23 ± 0.07	1165	1162
45	Nonanal	0.34 ± 0.11	1235	2.57 ± 0.17	1268	1267
46	Decanal			0.28 ± 0.02	1367	1366
47	Dodecanal			0.28 ± 0.10	1663	
48	α-Hexylcinnamaldehyde	5.46 ± 0.08		0.11 ± 0.12	1719	1728
49	Heptadecanal			1.18 ± 0.33	2088	
*Ether*				*0.09*		
50	Benzyl methyl ether			0.09 ± 0.01		966
*Terpene*		0.54		0.08		
51	*p*-Cymene			0.08 ± 0.04	1018	1025
52	α-Ionone	0.36 ± 0.15	1415			1413
53	β-Ionone	0.18 ± 0.02	1472			1486
Ketones				0.54		
54	Benzophenone			0.09 ± 0.02	1600	1611
55	2-Nonadecanone			0.45 ± 0.08	2116	2087
*Alcohols*		*12.15*		*0.62*		
56	Phenol			0.31 ± 0.09	1045	1043
57	2-Ethylhexanol			0.22 ± 0.04	1095	
58	1-Dodecanol	4.31 ± 0.00	1559	0.09 ± 0.01	1559	1575
59	1-Tridecanol	0.42 ± 0.00	1659			1656
60	1-Tetradecanol	3.43 ± 0.00	1765			1770
61	1-Hexadecanol	3.47 ± 0.00	1977			1965
62	1-Octadecanol	0.52 ± 0.00	2175			2159
*Alkene*				*0.42*		
63	1-Tridecene			0.42 ± 0.00	1284	1287
*Alkane*		*1.80*		*2.41*		
64	Hexadecane	1.80 ± 0.00	1581	1.11 ± 0.29	1581	1600
65	Octadecane			0.25 ± 0.00	1797	1800
66	Heneicosane			1.05 ± 0.21	2309	
	**Total**	**79.57**		**90.13**		

RA, relative abundance; KI Exp, Kovats index experimental; KI Ref, Kovats index reference.

**Table 2 plants-13-00963-t002:** Volatilomes of *Opuntia ficus-indica* (OFI) cultivars before and after infestation by *Dactylopius* species.

No.	Compounds	OFI Esmeralda	OFI Esmeralda Infested by *D. coccus*	KIExp	OFI Rojo Pelón	OFI Rojo Pelón Infested by *D. opuntiae*	KI Exp	KI Ref
		RA (%)	RA (%)		RA (%)	RA (%)		
	*Carboxilic acid and derivatives*	48.79	44.28		31.78	20.05		
1	Hexanoic acid	0.83 ± 0.63	0.82 ± 0.00	942				904
67	3-Methyl-2-pentenoic acid	0.35 ± 0.32		959				926
68	2-Hexenoic acid	0.66 ± 0.01	0.36 ± 0.00	972				939
69	4-Oxopentanoic acid	0.22 ± 0.14		991				956
70	Heptanoic acid	0.59 ± 0.50	0.39 ± 0.05	1022	0.39 ± 0.11	0.25 ± 0.15	1030	1005
5	2-Ethylhexanoic acid				0.11 ± 0.07		1044	1031
71	4-Methylvaleric acid		0.15 ± 0.10	1033				1039
72	2-Methyl-4-pentenoic acid		0.11 ± 0.16	1062				
73	Lactic acid	3.22 ± 1.91	1.63 ± 0.16	1065				1057
15	Benzoic acid	1.49 ± 1.77	0.29 ± 0.03	1080	2.06 ± 0.13	0.50 ± 0.09	1080	1084
74	Methyl benzoate	0.41 ± 0.14		1081		0.91 ± 0.34		1084
9	Octanoic acid	1.23 ± 1.52		1112	1.21 ± 0.07	1.34 ± 0.07	1112	1109
11	Ethyl Benzoate	0.05 ± 0.01		1156	0.08 ± 0.04	0.24 ± 0.11	1152	1141
75	Benzeneacetic acid					0.17 ± 0.05	1160	1150
76	Salicylic acid					0.18 ± 0.25	1171	1176
77	Methyl salicylate	1.21 ± 0.37	6.96 ± 0.14	1181		0.32 ± 0.07	1172	1176
78	2-Nonenoic acid		0.86 ± 0.00	1179		0.33 ± 0.08	1184	
13	Nonanoic acid	1.09 ± 0.24	0.76 ± 0.07	1216	1.45 ± 0.12	1.72 ± 0.14	1212	1206
79	Ethyl salycilate		1.03 ± 0.01	1244				1241
18	Decanoic acid	0.77 ± 0.75	0.63 ± 0.00	1318	0.79 ± 0.08		1311	1309
19	Butanedioic acid	0.49 ± 0.19		1320				1314
80	Gliceric acid	0.73 ± 0.02		1346				1342
81	2-Methoxybenzoic acid		0.10 ± 0.01	1331				1362
82	Methyl 2-methoxy benzoate		0.45 ± 0.10	1319				1295
83	Glutaric acid	0.16 ± 0.03		1410				1400
22	Undecanoic acid	0.16 ± 0.02	0.14 ± 0.07	1417	0.08 ± 0.02	0.12 ± 0.08	1408	1410
24	Dodecanoic acid	5.70 ± 6.08	2.77 ± 0.06	1516	7.19 ± 0.50	2.35 ± 0.16	1505	1509
84	2,5-Dimethoxy benzenemethanol acetate				0.13 ± 0.04		1523	
26	Ethyl Dodecanoate	0.26 ± 1.27	0.17 ± 0.09	1582	0.33 ± 0.01		1571	1581
28	*p*-Hydroxybenzoic acid	0.94 ± 0.45		1620				1621
29	Hexylsalicylate	0.25 ± 3.04	0.46 ± 0.07	1662				1652
85	Methyl tetradecanoate	0.84 ± 0.16		1719				1714
27	Tridecanoic acid		0.06 ± 0.04	1611				1606
86	12-Methyltridecanoic acid		0.07 ± 0.00	1678				1680
31	Tetradecanoic acid	2.98 ± 3.09	1.36 ± 0.06	1720	1.56 ± 0.10	0.19 ± 0.07	1714	1714
87	Methyl benzoate		1.46 ± 0.03	1752				
88	Benzyl Benzoate		2.65 ± 0.51	1741	0.15 ± 0.04		1754	1765
32	Ethyl tetradecanoate		0.10 ± 0.13	1784				1782
89	Nonanedioic acid	0.28 ± 0.01		1808				1788
90	Pentadecanoic acid	0.42 ± 0.47	0.43 ± 0.02	1826				1807
91	Isopropyl tetradecanoate		0.05 ± 0.34	1820				1827
92	Benzyl salicylate		0.31 ± 0.12	1855				1860
34	Hexadecanoic acid	7.19 ± 2.16	4.88 ± 0.16	1935	9.35 ± 0.74	8.25 ± 0.45	1916	1909
93	15-Methylhexadecanoic acid	0.17 ± 0.03		2040				1974
37	(Z,Z)-9,12-Octadecadienoic acid	1.90± 0.44	1.03 ± 0.12	2106	0.88 ± 0.14	2.20 ± 1.43	2087	2087
38	(Z)-9-Octadecenoic acid	2.35 ± 1.66	1.94 ± 0.10	2113	2.47 ± 0.98		2090	2100
39	Octadecanoic acid	2.53 ± 0.59	1.76 ± 0.01	2141	2.76 ± 0.15	0.59 ± 0.15	2119	2133
94	Methyl octadecanoate				0.33 ± 0.04		1809	
40	Ethyl octadecanoate		0.11 ± 0.00	2199				2202
41	Dehydroabietic acid	9.32 ± 2.24	9.99 ± 0.04	2376	0.46 ± 0.05	0.39 ± 0.02	2344	2373
	*Aldehides and derivatives*	2.15	6.25		4.3	4.82		
42	Hexanal	0.44 ± 0.44	0.47 ± 0.00	984				964
43	Heptanal		0.18 ± 0.02	1069				1068
95	Benzaldehyde		0.15 ± 0.07	1094	0.32 ± 0.03	0.55 ± 0.33	1094	1080
96	Diethyl acetal hexanal	0.25 ± 0.14	0.46 ± 0.10	1086				1082
97	5,5-Dimethyl-3-oxo-1-cyclohexene-1-carboxaldehyde		0.15 ± 0.03	1104				
44	Octanal		0.17 ± 0.01	1160	0.40 ± 0.05	0.09 ± 0.05	1165	1167
98	Phenylacetaldehyde	0.62 ± 0.61	0.56 ± 0.06	1198	0.82 ± 0.06	0.81 ± 0.17	1201	1208
45	Nonanal	0.53 ± 0.21	1.03 ± 0.01	1271	1.73 ± 0.12	1.32 ± 0.09	1265	1267
46	Decanal		0.14 ± 0.00	1370	0.14 ± 0.08		1366	1366
99	Nonanaldimethylacetal	0.21 ± 0.10	0.37 ± 0.05	1374				1379
100	3-(4-(*tert*-butyl)phenyl-2-methylpropanal		0.30 ± 0.04	1497				1500
101	4-Hydroxy-3-methoxybenzaldehyde	0.10 ± 0.04	0.62 ± 0.02	1524	0.89 ± 0.04	2.05 ± 0.01	1511	1544
102	3-Ethoxy-4-hydroxybenzaldehyde		0.11 ± 0.02	1554				1560
48	α-Hexylcinnamaldehyde		1.22 ± 0.74	1725				1726
103	Octadecanal		0.32 ± 0.13	2187				
	*Heterocycles*		0.67		8.91	1.38		
104	2-Isopropyl-3-metoxypirazina		0.25 ± 3.25	1070				1080
105	2-Methoxy-3-isopropylpyrazine				0.30 ± 7.22	1.05 ± 0.33	1083	1089
106	Ethyl 2-(5-methyl-5-vinyltetrahydrofuran-2-yl)propan-2-yl carbonate				8.18 ± 0.10		1064	1090
107	3-Isobutyl-2-methoxypyrazine				0.43 ± 0.03		1164	1170
108	3-Ethyl-4-methyl-1H-pyrrole-2,5-dione		0.35 ± 0.03	1209				1192
109	3-Hydroxy-2-methylpyran-4-one		0.07 ± 0.00	1266				1293
110	2,3-Dihydro-2,2,4,6-tetramethylbenzofuran					0.33 ± 0.01	1410	
	*Ethers*	0.31	0.16					
50	Benzylmethylether	0.31 ± 0.20		966				966
111	1,2-Dimethoxybenzene		0.16 ± 0.02	1111				1106
	*Ketones*	1.95	1.16		1.17	0.61		
112	5-Hexen-2-one	0.29 ± 0.01		1007				
113	2,2,6-Trimethylcyclohexanone				0.14 ± 0.09		1031	
114	Acetophenone		0.57 ± 0.08	1047		0.45 ± 0.05	1055	1049
115	Isophorone	0.24 ± 0.06	0.01 ± 0.22	1106	0.17 ± 0.03		1038	1094
116	Phenylacetone	0.48 ± 0.05	0.44 ± 0.00	1114	0.24 ± 0.02	0.16 ± 0.07	1110	1116
117	4-Oxoisophorone	0.13 ± 0.02		1131	0.07 ± 0.04		1125	1105
118	2-(1-Hydroxybut-2-enylidene)cyclohexanone				0.14 ± 0.03		1145	
119	1-(1-cyclohexen-1-yl)(-1-1-Butenone)	0.70 ± 0.34		1214				
54	Benzophenone	0.11 ± 0.07	0.14 ± 0.01	1607	0.41 ± 0.05		1584	1607
	*Terpenes*	17.89	13.92		0.8	15.52		
120	Limonene	0.85 ± 0.23		1023				1020
121	Linalool oxide	8.48 ± 0.58	5.06 ± 0.42	1063				1064
122	*trans*-Linalool oxide		5.70 ± 0.03	1064				1068
123	1,5,5-Trimethyl-3-methylene cyclohexene	0.55 ± 0.63		1071				
124	β-Linalool	5.00 ± 0.58	0.26 ± 0.35	1088				1082
125	α-Terpineol	2.00 ± 0.36		1178				1172
126	Linalool	0.19 ± 0.22		1232		5.61 ± 0.10	1227	1227
127	Geraniol	0.44 ± 0.27	0.33 ± 0.04	1250		1.84 ± 0.06	1357	1238
128	Nerol		0.33 ± 0.06	1232		0.79 ± 0.04	1328	1260
129	β-Damascenone		0.18 ± 0.17	1362		0.55 ± 0.30	1360	1361
52	α-Ionona		0.09 ± 0.26	1404				1413
130	α-Isomethylionone		0.88 ± 0.15	1453				1478
53	β-Ionone		0.40 ± 0.06	1460		0.18 ± 0.04	1458	1486
131	Dihydroactinidiolide	0.38 ± 0.13	0.30 ± 0.02	1537				1532
132	Neophytadiene		0.39 ± 0.08	1832				1842
133	28-Nor-17β(H)-hopane				0.45 ± 0.01		2942	
134	β-Sitosterol				0.35 ± 0.03	6.55 ± 0.04	3244	3284
	*Alcohols*	12.72	9.91		29.37	30.78		
135	1,2-Dihydroxy-4-methylpentane		0.27 ± 0.02	990				
136	Hexanol		0.06 ± 0.01	9994				992
137	(*Z*)-2-Hexen-1-ol		0.36 ± 0.19	1010	7.65 ± 0.18	4.21 ± 0.08	1025	1001
56	Phenol	0.22 ± 0.09	0.16 ± 1.30	1045				1043
138	Heptanol		0.11 ± 0.23	1067				1092
57	2-Ethylhexanol	0.58 ± 0.23	0.22 ± 0.24	1099	2.13 ± 0.09	1.49 ± 0.05	1103	
139	Benzylalcohol	0.27 ± 0.11	0.56 ± 0.22	1143	0.90 ± 0.02	2.08 ± 0.06	1132	1156
140	1-Octanol		0.29 ± 0.14	1158	1.27 ± 0.04	1.51 ± 0.18	1177	1177
141	Guaiacol		0.35 ± 0.03	1209				1192
142	Nonanol		0.07 ± 0.01					
143	Glycerol	0.33 ± 0.52		1290	1.46 ± 0.09		1288	1292
144	*p*-Vinylguaiacol	10.62 ± 7.24	1.76 ± 0.34	1305	14.67 ± 0.93	17.03 ± 5.14	1294	1282
145	1-Methyl-1(4-methyl-3-cyclohexenyl)ethanol		0.63 ± 0.00	1318				1309
146	Isododecanol		0.09 ± 0.01	1479				
58	1-Dodecanol	0.47 ± 0.09	0.60 ± 0.01	1563	0.76 ± 0.04		1553	1575
60	1-Tetradecanol		0.66 ± 0.00	1756				1768
61	1-Hexadecanol	0.23 ± 0.07	0.98 ± 0.08	1978	0.53 ± 0.03		1960	1965
62	1-Octadecanol		0.92 ± 0.04	2177				2159
147	3,7,11,15-Tetramethyl-2-hexadecenol		1.82 ± 0.15	2198		3.52 ± 0.17	2173	2179
148	Octacosanol					0.94 ± 0.04	3125	3154
	*Aromatic derivatives*	0.77						
149	1,2-Dihydro-1,1,6-trimethyl naphthalene	0.18 ± 0.07		1338				1332
150	10,18,Bisnorabieta-8,11,13.triene	0.59 ± 0.03		2041				
	*Alkanes*	0.69	0.55		0.78			
64	Hexadecane	0.69 ± 0.19	0.07 ± 0.01	1585				1600
151	Heptadecane		0.12 ± 0.05	1692				1700
152	Nonadecane		0.36 ± 0.06	1906				1900
153	Eicosane				0.19 ± 2.08		1992	2000
66	Heneicosane				0.23 ± 0.42		2092	2100
154	Docosane				0.36 ± 0.07		2188	2200
**Total**		**85.27**	**76.9**		**77.11**	**73.16**		

RA, relative abundance; KI Exp, Kovats index experimental; KI Ref, Kovats index reference.

## Data Availability

The datasets used and analyzed during the current study are available from the corresponding author upon reasonable request.

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
