# Peer review of "Interactions of Opuntia ficus-indica with Dactylopius coccus and D. opuntiae (Hemiptera: Dactylopiidae) through the Study of Their Volatile Compounds"

_plants, 2024, doi:10.3390/plants13070963_

Round 1

Reviewer 1 Report

Comments and Suggestions for Authors

L 23, 24 and others. In general, I understand that it is convenient to use the more defined term BVOCS rather than VOCS, although given the profile of the work, either term is acceptable.

L 24. to determine the VOCs produced by D. coccus and D. opuntiae… 

L72-73. This sentence needs References

L 185-193. This paragraph should be better placed in Methods 4.3

L 366-374. It would be convenient to focus on identifying the percentage contributions of each volatile compound to the average dissimilarity groups of plants. For this, SIMPER (the simper function in vegan) (Clarke KR (1993) Non-parametric multivariate analyses of changes in community structure. Aust J Ecol 18: 117–143. doi.wiley.com/10.1111/j.1442-9993.1993.tb00438.x 413414; Clarke KR, Gorley RN, Somerfield PJ, Warwick RM (2014) Change in marine communities: an approach to statistical analysis and interpretation, 3nd edition. PRIMER-416 E Ltd, Plymouth, 1-262) or similar alternatives can be used

Reviewer 2 Report

Comments and Suggestions for Authors

The manuscript is well written apart from the methodology which is too brief and as such does not describe the experiments done sufficiently.  

For example, I could not tell how the separation of insect and plants VOCs was achieved.  

Below are additional suggestions.

Line 42; Consider replacing “food specificity” with “host specificity” .

Line 86; objective 1 to determine the VOCs 86 of D. coccus and D. opuntiae feeding on O. ficus-indica.  

Based on what you’ve presented in the methodology, its not clear to me how you collected and separated VOCs of insects from their host plant volatiles. 

Line 102 to 103 Line 96-108

As above, I am sorry if I am missing some thing obvious. how were you able to separate VOCs from insects and plants? Or are these volatiles from D. coccus and D. opuntiae infested plants?

Materials and methods

How were the samples prepared for hydrodistillation? If plants were damaged during sample preparations, how were you able to separate the effects of biotic and abiotic stress on production of VOCs?

Line 326: One hundred grams of adult female D. coccus and D. opuntiae and 1000 g of infested 326 and uninfested O. ficus indica cladodes were hydrodistilled.

Were insects and their host plants kept together during hydrodistillation? Please clarify.

Can you clarify if one hundred grams was the combined wight of D. coccus and D. opuntiae or of each species.

Same as above is 1000 g the combined wight of infested and uninfested O. ficus-indica cladodes.

Reviewer 3 Report

Comments and Suggestions for Authors

Plants

Interactions of Opuntia ficus-indica with Dactylopius coccus and D. opuntiae (Hemiptera: Dactylopiidae) through the study of their volatile compounds

This manuscript reports the collection and identification of VOCs (through hydrodistillation) produced by two species of scale insects, D. coccus and D. opuntiae, and the cactus O. ficus-indica. VOCs profiles were compared between plants that were uninfected and those that were infested with D. coccus and D. opuntiae.

Abstract

L 19 – 21: This sentence is confusing and should be rewritten for clarity.

Materials and Methods

L 326: more information should be included about the distillation method, for example how old were the insects used in the distillation?

Comments on the Quality of English Language

N/A
